# Circulating IL-6, IL-10, and TNF-alpha and IL-10/IL-6 and IL-10/TNF-alpha ratio profiles of polyparasitized individuals in rural and urban areas of gabon

Noé Patrick M'Bondoukwé[1]*, Reinne Moutongo[1], Komi Gbédandé[2], Jacques Mari Ndong Ngomo[1], Tatiana Hountohotegbé[2], Rafiou Adamou[2], Jeanne Vanessa Koumba Lengongo[1], Kowir Pambou Bello[3], Denise Patricia Mawili-Mboumba[1], Adrian John Frederick Luty[4], Marielle Karine Bouyou-Akotet[1]

**1** Department of Parasitology-Mycology, Faculty of Medicine, University of Health Sciences, Libreville, Gabon, **2** Centre d'Etude et de Recherche sur le Paludisme Associé à la Grossesse et à l'Enfance (CERPAGE), Cotonou, Benin, **3** Ecole Normale Supérieure (E.N.S.), Libreville, Gabon, **4** Université de Paris, MERIT, IRD, Paris, France

* mbondoukwenoe@gmail.com

## Abstract

Malaria, blood-borne filarial worms and intestinal parasites are all endemic in Gabon. This geographical co-distribution leads to polyparasitism and, consequently, the possibility of immune-mediated interactions among different parasite species. Intestinal protozoa and helminths could modulate antimalarial immunity, for example, thereby potentially increasing or reducing susceptibility to malaria. The aim of the study was to compare the cytokine levels and cytokine ratios according to parasitic profiles of the population to determine the potential role of co-endemic parasites in the malaria susceptibility of populations. Blood and stool samples were collected during cross-sectional surveys in five provinces of Gabon. Parasitological diagnosis was performed to detect plasmodial parasites, *Loa loa*, *Mansonella perstans*, intestinal helminths (STHs) and protozoan parasites. Nested PCR was used to detect submicroscopic plasmodial infection in individuals with negative blood smears. A cytometric bead array was used to quantify interleukin (IL)-6, IL-10 and tumour necrosis factor (TNF)-α in the plasma of subjects with different parasitological profiles. Median IL-6 and IL-10 levels and the median IL-10/TNF-α ratio were all significantly higher among individuals with *Plasmodium* (*P.*) *falciparum* infection than among other participants ($p<0.0001$). The median TNF-α level and IL-10/IL-6 ratio were higher in subjects with STHs ($p = 0.09$) and *P. falciparum*-intestinal protozoa co-infection ($p = 0.04$), respectively. IL-6 (r = -0.37; $P<0.01$) and IL-10 (r = -0.37; $P<0.01$) levels and the IL-10/TNF-α ratio (r = -0.36; $P<0.01$) correlated negatively with age. Among children under five years old, the IL-10/TNF-α and IL-10/IL-6 ratios were higher in those with intestinal protozoan infections than in uninfected children. The IL-10/TNF-α ratio was also higher in children aged 5–15 years and in adults harbouring blood-borne filariae than in their control counterparts, whereas the IL-10/IL-6 ratio was lower in those aged 5–15 years with filariae and intestinal parasites but higher in adults with intestinal parasitic infections. Asymptomatic malaria is associated with a strong polarization

**Data Availability Statement:** All relevant data are within the manuscript and its Supporting Information files.

**Funding:** The Department of Parasitology-Mycology of Tropical Medicine, the Gabonese Red Cross, the CEGADIS group and the RELACS network are the organizations that funded the project. All funds have been invested in the Ph.D. thesis of NPM. The Department of Parasitology-Mycology of Tropical Medicine allowed to NPM to have salary and to research other source of funding. The Gabonese Red Cross worked with the Department of Parasitology-Mycology of Tropical Medicine to collect data and samples in rural areas of Gabon. The Gabonese Red Cross supported this project providing transport in the field and material for parasitosis diagnosis. The CEGADIS group has been solicited for travel-related expenses in Benin where has been carried out immunological analysis. To finish the RELACS network allowed attending to training workshops. These funders, except The Department of Parasitology-Mycology of Tropical Medicine which is my institution, had no role in study design, data collection and analysis, decision to publish, or preparation of the manuscript.

**Competing interests:** The authors have declared that no competing interests exist.

towards a regulatory immune response, presenting high circulating levels of IL-10. *P. falciparum*/intestinal protozoa co-infections were associated with an enhanced IL-10 response. Immunity against malaria could differ according to age and carriage of other parasites. Helminths and intestinal protozoa can play a role in the high susceptibility to malaria currently observed in some areas of Gabon, but further investigations are necessary.

## Author summary

The current epidemiological transition of malaria observed in Gabon included, for example, a shift in the at-risk population from children aged less than 5 years old to older children aged 5–15 years. Another consequence was the increasing number of cases of infection among adults. In view of these findings, it is important to explain this phenomenon of epidemiological modification of malaria in Gabon. Intestinal parasites and blood filariasis are endemic in Gabon. These parasites are described to alter the malaria immune response and can be implicated in the susceptibility of individuals to malaria. In Gabon, malaria presents a heterogeneous repartition. Thus, in the present study, we investigated the role of co-endemic parasitosis in the alteration of the malarial immune response by comparing Th1 (IL-6 and TNF-α) and Th2/Treg (IL-10) cytokine production between mono- and co-parasitized individuals in many localities with different epidemiological patterns of malaria. Microscopic analyses and rapid antigenic tests were performed for malaria diagnosis. The nested PCR technique was used to demonstrate the submicroscopic parasitaemia of *Plasmodium* sp. Then, once groups with different parasitological profiles were constituted, IL-6, IL-10 and TNF-α levels were measured in the plasma of individuals. Th2/Th1 ratios, which can indicate the level of susceptibility of individuals to malaria, were calculated. We observed that there was no interaction between *Plasmodium* sp. and co-endemic parasites in the present study. However, the high Th2/Th1 cytokine ratio among patients with intestinal protozoa seems to suggest that these intestinal parasites could also play a role in susceptibility to malaria as they do for helminths.

## Introduction

In Gabon, malaria and intestinal parasite infections (IPIs) as well as filariasis are frequently diagnosed in the population, with the prevalence varying from 1% to 75% [1–4]. This geographical co-distribution can lead to polyparasitism with possible interactions between parasite species. Studies on the association between *Plasmodium* and helminths have shown that helminths could have either a protective, a detrimental or a neutral effect on plasmodial infections. For example, in Mali, *Schistosoma* (*S.*) *haematobium* delayed the appearance of clinical malaria in children, while in Gabon and in Senegal, it was reported that *Ascaris lumbricoides* and *S. haematobium* increased the risk of clinical episodes [5–7]. Shapiro and colleagues found no interactions between these STHs and malaria parasites [8]. Although the epidemiological patterns and/or the clinical consequences of polyparasitism are being increasingly studied, data concerning the immune responses and the susceptibility to other diseases of individuals exposed to or infected by different helminths or protozoal parasites in endemic areas remain scarce. Relevant information for Gabon does not exist. Understanding the immune response elicited by each pathogen in the case of co-infection could help with the management or prevention of the deleterious effects of polyparasitism.

Levels of pro- and anti-inflammatory cytokines that are influenced either by the environment or by other parasites can reflect the *P. falciparum*-induced immunity and morbidity of populations living in regions with different malaria endemicities. Indeed, interleukin (IL)-10 was shown to be a predictor of the occurrence of clinical malaria in highly endemic areas of India, while IL-6, IL-10 and IL-12 were associated with disease outcome in a non-endemic region [9]. With respect to *Plasmodium*-helminth co-infections, helminths orient the immune response towards an anti-inflammatory pathway, while plasmodial infection is known to elicit a strong pro-inflammatory response [10,11]; filaria-helminth co-infection is associated with a low IL-10 level, whereas that of IL-6 was described to be lower in the case of *Plasmodium*-hookworm co-infection [12].

On the other hand, chronic intestinal helminthic and/or filarial infection is associated with high levels of IL-10 [13]. IL-10 downregulates the functions of immune cells that release pro-inflammatory cytokines, both preventing the elimination of worms and protecting the host against helminth infection-related symptoms. IL-6 levels are also described in the literature to increase significantly among populations infected with helminths [14]. However, the level of TNF-α is reported to not vary [14]. Thus, intestinal parasite and filarial carriage could influence the susceptibility to and/or the course of malaria, which is known to differ between urban and rural settings in high transmission areas. As an example, in a rural setting, antimalarial immunity was shown to be modulated by infection with *Trichuris trichiura* [15]. Nevertheless, in urban and suburban areas, data on such interactions are lacking. The heterogeneity of *Plasmodium spp.* prevalence observed in Gabon may be due to different susceptibility patterns to *P. falciparum* infection according to the presence and type of other parasitoses. The study presented here was conducted in areas of Gabon not yet investigated that have different levels of malaria endemicity. In the country, among intestinal parasitisms, compared to STHs, protozoans predominate in single or in co-infection with *P. falciparum* [16,17], but the interaction between such parasite species carriage and *P. falciparum*-related immunity has not been studied, as non-pathogenic protozoan carriage does not constitute a public health problem. However, experimental studies on *Entamoeba (E.) histolytica* and *Giardia (G.) intestinalis* have demonstrated that parasite carriage is associated with the production of pro-inflammatory cytokines [18,19]. Subtypes of *Blastocystis* spp., whose prevalence is increasing in the country, are now considered pathogenic intestinal protozoa. In humans, this parasite was also associated with immunomodulation, precisely by downregulating the immune response towards another antigen such as helminths [18,19]. This study evaluated the impact of intestinal parasites (helminths or protozoa) and blood filaria carriage on the level of cytokines involved in malaria immunity during co-infection with helminths and protozoa in urban and rural areas of Gabon.

## Results

### Patients and parasites

Samples of 240 participants were selected for the immunological analysis (Fig 1). Out of the 208 patients for whom the area was known, 172 (82.7%) were from a rural area (Table 1). Age was recorded for 224 of participants, and the median age was 22.5 [6.0–48.8] years old. Adults represented more than half of the study population (58.0%; 130/224), and children less than 5 years old represented 21.0% (47/224). The sex ratio (number of males/number of females) was 0.83 (109/131) and did not differ according to age or site.

Combined microscopic and molecular diagnoses showed the prevalence of single *Plasmodium* infection to be 20.8% (50/240). *P. falciparum* was the only species identified. Intestinal helminth (STH) and filariasis infections were detected in 43 (17.9%) and 48 (20.0%)

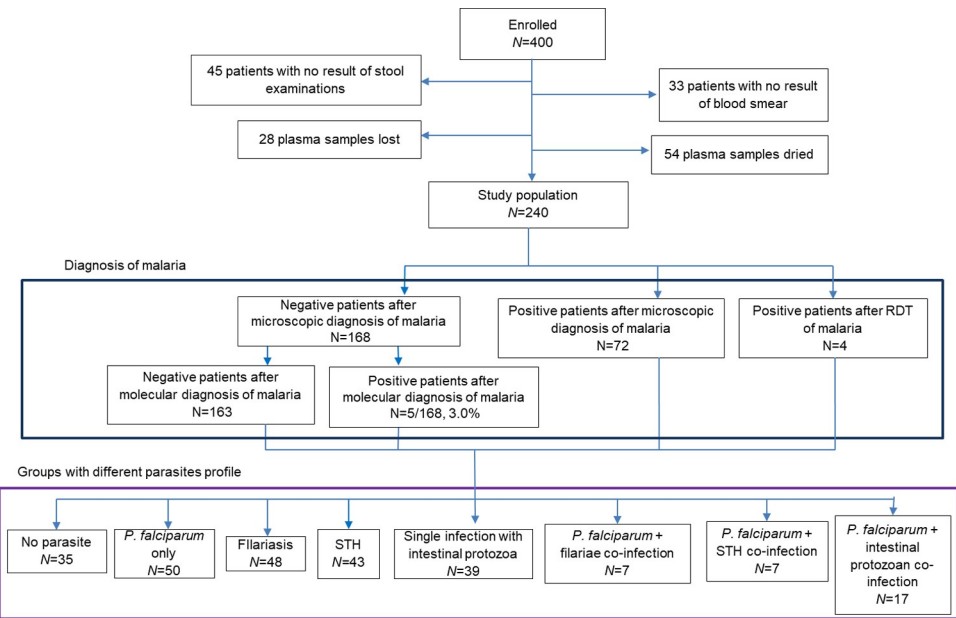

**Fig 1. Flow chart of the study population.** This figure represents a flow chart for enrolment of the study population, followed by the number of individuals found with malarial infections in the study population and absolute frequencies in each group with different parasitic profiles.

individuals, respectively, and intestinal protozoan monoinfection was detected in 39 (16.3%) individuals. *A. lumbricoides* (14.6%; 35/240), *T. trichiura* (13.7%; 33/240), *B. hominis* (36.2%; 87/240), and *E. histolytica/dispar* (7.1%; 17/240) were the most frequently detected intestinal parasites; *G. duodenalis* and *S. stercoralis* were each found in only 5/240 (2.1%) samples. *Schistosoma* species were not detected in any of the samples. The rates of *P. falciparum*/STH and *P. falciparum*/filariae co-infections were 2.9% (7/240), and that of *P. falciparum*/intestinal protozoan co-infection was 7.1% (17/240) (Table 1).

The prevalence of parasitic infection of any type was highest in adults (87.7%; 114/130), followed by those aged 5-to-15 years old (87.2%; 41/47), and lowest in children less than 5 years

**Table 1. Distribution of the groups with different parasitic profiles according to sex, age and location.**

| | Total | No parasites (n = 35) | | P. falciparum only (n = 50) | | Filariae (n = 48) | | STH (n = 43) | | Intestinal protozoa only (n = 39) | | Plasmodium/ filariae co-infection (n = 7) | | Plasmodium/ STH co-infection (n = 7) | | Plasmodium/ intestinal protozoa co-infection (n = 17) | | P |
|---|---|---|---|---|---|---|---|---|---|---|---|---|---|---|---|---|---|---|
| | N | n | % | n | % | n | % | n | % | n | % | n | % | n | % | n | % | |
| Male sex | **109** | 17 | 48.6 | 24 | 48.0 | 21 | 47.7 | 11 | 26.8 | 20 | 51.3 | 3 | 42.9 | 4 | 57.1 | 9 | 52.9 | 0.4 |
| Age groups | **224** | **35** | | **49** | | **43** | | **34** | | **39** | | **7** | | **5** | | **17** | | <0.0001 |
| 0–4 years old | 47 | 13 | 37.2 | 16 | 32.6 | 0 | 0.0 | 4 | 11.8 | 9 | 23.1 | 0 | 0.0 | 3 | 50.0 | 2 | 11.8 | |
| 5–15 years old | 47 | 6 | 17.1 | 19 | 38.8 | 1 | 2.3 | 6 | 17.7 | 11 | 28.2 | 0 | 0.0 | 1 | 16.7 | 9 | 52.9 | |
| >15 years old | 130 | 16 | 45.7 | 14 | 28.6 | 42 | 97.7 | 24 | 70.5 | 19 | 48.7 | 7 | 100.0 | 2 | 33.3 | 6 | 35.3 | |
| Location | **208** | **30** | | **49** | | **41** | | **37** | | **27** | | **7** | | **7** | | **10** | | <0.0001 |
| Urbanized area | 36 | 12 | 40.0 | 15 | 30.6 | 0 | 0.0 | 0 | 0.0 | 6 | 22.2 | 0 | 0.0 | 0 | 0.0 | 3 | 30.0 | |
| Rural area | 172 | 18 | 60.0 | 34 | 69.4 | 41 | 100.0 | 37 | 100.0 | 21 | 77.8 | 7 | 100.0 | 7 | 100.0 | 7 | 70.0 | |

*p* values were obtained using Fisher's exact test.

old (72.3%; 34/47) ($\chi^2$ = 33.932, $df$ = 2, $P$<0.0001). Adults were more frequently infected than children (Table 1). The prevalence of filariae (97.7%), STH (70.5%) and intestinal protozoa (48.7%) was significantly higher among adults than among children ($\chi^2$ = 71.747, $df$ = 14, $P$<0.0001) (Table 1). Moreover, all those co-infected with *P. falciparum*/filariae were adults. The proportions of individuals with *P. falciparum*/STH co-infections and *P. falciparum*/intestinal protozoan co-infections were higher in younger children aged less than 5 years old (50.0%) and those aged between 5 and 15 years old (52.9%).

P values presented in this paragraph were considered globally for comparison of the parasitaemia and filaraemia between groups. To explore the significance in more detail, see S1 Table. The median *P. falciparum* parasitaemia density was significantly higher in cases of monoinfection (9450 [1296–36400] T/µL ($P$≤0.04), while it was lower in those with either STH (42 [12.3–681.5] T/µL), filarial (350 [16–1400] T/µL) or intestinal protozoan (749 [35–10150] T/µL) co-infections ($P$≤0.04). *L. loa* and *M. perstans* parasite densities were comparable between groups. Intestinal protozoa were found to be more frequently associated with other parasites (28.7%; 69/240).

## Cytokine levels in uninfected individuals

The median IL-6, IL-10 and TNF-α levels were 14.2 [4.7–68.0] pg/mL, 11.1 [8.0–15.2] pg/mL and 5.4 [4.6–6.8] pg/mL, respectively. The IL-10/TNF-α ratio was 2.0 [1.4–3.2] and that of IL-10/IL-6 was 1.2 [0.2–2.3]. The "no parasite" group was used as a reference to estimate increasing, decreasing and comparable cytokine levels and cytokine ratios (Table 2).

## Cytokine levels in mono- and polyparasitized individuals

The applied Kruskal–Wallis test showed trends towards differences (p<0.05) in the cytokine levels and cytokine ratios except for those of TNF-α (p = 0.09) (Table 2). The descriptive

**Table 2. Median levels of pro- and anti-inflammatory cytokines according to parasitic profile.**

| Cytokines | Median [25$^{th}$ - 75$^{th}$ percentiles] (pg/mL) | | | | | | | | | |
|---|---|---|---|---|---|---|---|---|---|---|
| | Global | No parasites | Malaria only | STH | Filariasis | Intestinal protozoan only | Malaria/filariasis co-infection | Malaria/STH co-infection | Malaria/intestinal protozoan co-infection | P |
| IL-6 | 8.5 [3.2–53.6] | 14.2 [4.7–68.0] | 124.5 [36.9–433.9] | 5.9 [0.8–9.1] | 7.9 [3.6–11.3] | 4.1 [0.8–8.2] | 7.4 [5.2–77.7] | 25.5 [4.4–60.8] | 7.5 [0.9–48.2] | <0.0001 |
| TNF-α | 5.3 [2.9–7.6] | 5.4 [4.6–6.8] | 5.8 [3.8–8.1] | 6.5 [3.5–11.7] | 4.6 [1.3–7.7] | 5.1 [3.3–5.6] | 3.9 [1.2–7.9] | 2.7 [1.7–5.8] | 1.3 [0.6–7.0] | 0.09 |
| IL-10 | 12.1 [6.9–59.6] | 11.1 [8.0–15.2] | 224.5 [78.0–657.9] | 8.0 [4.7–12.7] | 6.9 [3.3–13.4] | 8.9 [6.8–14.5] | 18.1 [12.8–125.2] | 39.5 [33.0–68.0] | 26.9 [8.7–60.1] | <0.0001 |
| IL-10/TNF-α | 2.3 [1.3–13.8] | 2.0 [1.4–3.2] | 69.9 [12.5–140.7] | 1.1 [0.7–1.9] | 1.9 [1.1–3.6] | 1.8 [1.3–4.4] | 19.7 [1.6–115.2] | 13.0 [3.5–24.9] | 16.9 [2.4–86.6] | <0.0001 |
| IL-10/IL-6 | 1.7 [0.9–4.4] | 1.2 [0.2–2.3] | 2.0 [0.9–3.6] | 1.7 [1.1–5.8] | 1.4 [0.6–2.5] | 2.1 [0.9–8.0] | 2.5 [1.8–4.7] | 1.5 [0.7–4.8] | 3.6 [2.0–11.9] | 0.04 |

Table 2 shows the median levels of cytokines and ratios in the 8 different parasitic groups. The Kruskal–Wallis H-test allowed the comparison of all groups together. *P* values of this comparison are presented in the last column of the table. The second column titled "No parasites" indicates the control group, which allows observation of an increase/decrease in cytokine levels compared to those in other groups. Values obtained by the Kruskal–Wallis H-test are presented on the right: **IL-6**: $\chi^2$ = 65.725, $df$ = 7, $P$<0.0001; **TNF-α**: $\chi^2$ = 11.107, $df$ = 7, $P$ = 0.09; **IL-10**: $\chi^2$ = 87.714, $df$ = 7, $P$<0.0001; **IL-10/TNF-α**: $\chi^2$ = 81.337, $df$ = 7, $P$<0.0001; **IL-10/IL-6**: $\chi^2$ = 14.781, $df$ = 7, $P$ = 0.04.

analysis of the cytokine measurements showed the following results. IL-6 median levels were higher among those with single *P. falciparum* (124.5 [36.9–433.9] pg/mL) and a *P. falciparum/* STH co-infection (25.5 [4.4–60.8] pg/mL). IL-10 content was also higher in participants with *P. falciparum* single infection (224.5 [78.0–657.9] pg/mL) and in *P. falciparum*-filariae co-infected (18.1 [12.8–125.2] pg/mL) and STH- (39.5 [33.0–68.0] pg/mL) and intestinal proto-zoa-infected participants (26.9 [8.7–60.1] pg/mL). STH-infected individuals had a higher median TNF-α level than uninfected individuals. The IL-10/TNF-α ratio was on average 30-fold (69.9 [12.5–140.7]), 10-fold (19.7 [1.6–115.2]), 6-fold (13.0 [3.5–24.9]) and 8-fold (16.9 [2.4–86.6]) higher in those with single or *P. falciparum* co-infections with filariae, STH and intestinal protozoa, respectively, than in uninfected participants. The IL-10/IL-6 ratio tended to be higher in those with *P. falciparum* infections alone (2.0 [0.9–3.6]) and 2-fold (2.5 [1.8–4.7]) and 3-fold (3.6 [2.0–11.9]) higher in those with dual infections with malaria/filariae and malaria/intestinal protozoans, respectively.

Fig 2A and 2B show pairwise comparisons of cytokine median concentrations and ratios. TNF-α levels did not differ significantly according to the presence or type of parasitic infection (Fig 2).

After Bonferroni correction in Fig 2A and considering a p value <0.000357, no significant relationship was found. Therefore, in the paragraph below, we presented only comparisons with p values nearest significance.

Subjects with plasmodial infection had only higher levels of IL-6 than subjects with *P. falciparum/*intestinal protozoa co-infection (*P* = 0.0008). In rural areas (Fig 2B), higher IL-10 concentrations were found in plasmodial-infected subjects than in those with *P. falciparum/* intestinal protozoa co-infection (*P* = 0.0007).

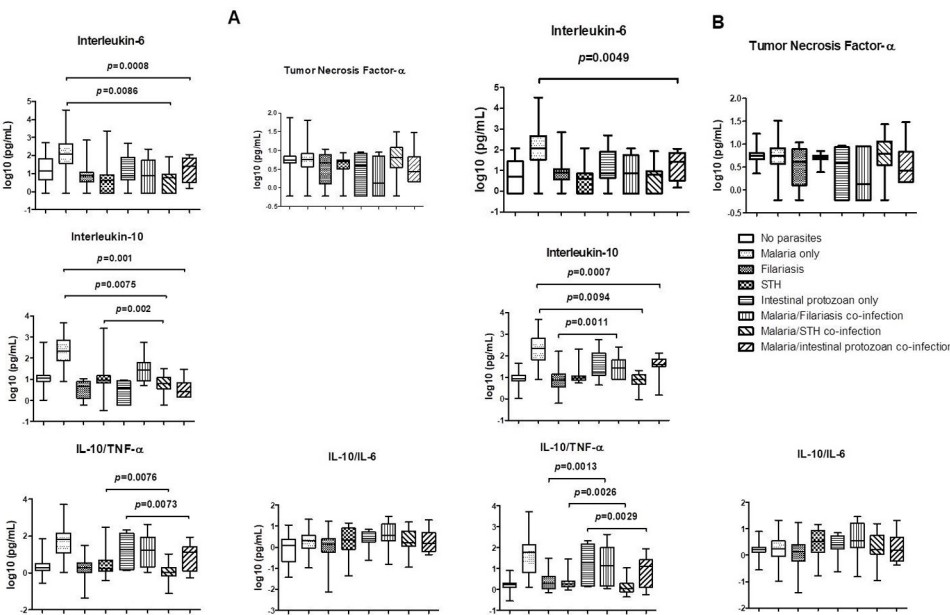

**Fig 2. Comparisons of cytokine concentrations between groups with different parasitic profiles.** The *p* value considered significant for analysis presented in this graphic is 0.000357. Only *p* values with two or more zero after the full stop of the decimal part were presented in the figure. All the associations showed no significant difference. (A) Box plot displaying IL-6, TNF-α and IL-10 cytokine production and IL-10/TNF-α and IL-10/IL-6 median ratios according to different parasitic profiles. The Mann–Whitney test was carried out for pairwise comparisons. Values used for the graphical representation were log-transformed. (B): Comparisons in rural areas only because there were not enough data for urban areas to perform this analysis.

## Cytokine profiles in relation to age

The levels of IL-6 and IL-10 and the IL-10/TNF-α ratio were inversely associated with the age of the study participants in the absence of parasitic infections (Fig 3 and Table 3). The same trend was observed in the case of *P. falciparum* single infection (for IL-6 only) for filariasis and intestinal protozoal infection (for all) (Table 3). Uninfected children aged 5–15 years old had the highest IL-10/IL-6 ratio (Table 3).

Overall, the IL-10 levels and IL-10/TNF-α ratios did not significantly vary between infected and uninfected adults. The median IL-6 level was between 3- and 4-fold lower in the case of a parasitic infection within the adult group, while an IPI was associated with a higher IL-10/IL-6 ratio.

Among children, a trend towards a lower TNF-α level in those aged below 5 years who had an STH carriage and a higher level in those aged between 5–15 years old was noted. IL-10 median levels were always lower in the case of STH regardless of age, whereas the level of IL-6 was reduced in individuals infected with filariae and intestinal parasitosis. This reduction was more pronounced for the youngest children with an STH.

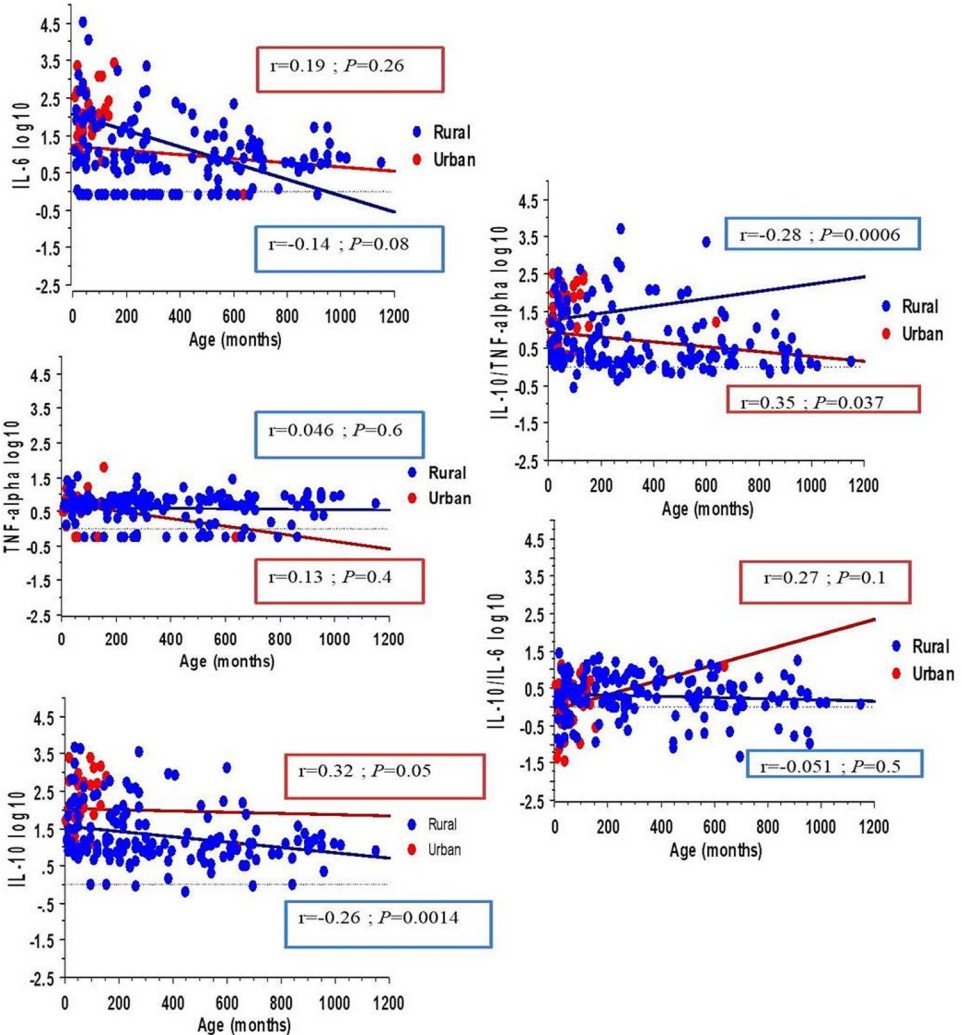

**Fig 3. Cytokines profiles stratified by areas according to age in months. IL-10 level decreased when age increased in rural area. IL-10/TNF-alpha ratio increased with age in rural area but decreased in urban area.**

**Table 3. Median level of cytokines according to single parasitism and age groups.**

|  |  | Median [25$^{th}$ - 75$^{th}$ interquartiles] | | |
|---|---|---|---|---|
| Groups |  | **0–4 years old** | **5–15 years old** | **> 15 years old** |
| No parasites | **IL-6** | 40.7 [12.2–124.1] | 30.1 [0.8–99.8] | 8.9 [4.1–39.4] |
|  | **TNF-α** | 5.3 [4.6–5.5] | 5.2 [4.5–6.5] | 6.2 [4.2–8.0] |
|  | **IL-10** | 13.1 [11.1–18.5] | 10.5 [6.9–223.8] | 9.6 [7.0–12.9] |
|  | **IL-10/TNF-α** | 2.4 [2.2–3.7] | 2.3 [1.2–25.4] | 1.6 [1.3–1.9] |
|  | **IL-10/IL-6** | 0.4 [0.1–1.4] | 5.0 [1.3–8.6] | 1.3 [0.6–3.2] |
| Malaria only | **IL-6** | 198.1 [64.4–515.7] | 137.8 [43.8–968.3] | 67.1 [23.0–213.1] |
|  | **TNF-α** | 5.0 [4.4–7.8] | 6.4 [5.1–8.1] | 5.6 [0.6–8.1] |
|  | **IL-10** | 180.1 [119.3–649.0] | 424.4 [59.5–755.4] | 196.2 [38.4–820.4] |
|  | **IL-10/TNF-α** | 60.9 [16.0–111.1] | 75.1 [8.0–132.5] | 80.7 [6.5–486.3] |
|  | **IL-10/IL-6** | 1.4 [0.7–2.8] | 2.1 [0.4–3.5] | 2.1 [1.0–5.5] |
| Filariasis | **IL-6** | - | 11.2 | 7.6 [3.3–11.0] |
|  | **TNF-α** | - | 9.3 | 4.6 [1.2–7.7] |
|  | **IL-10** | - | 32.1 | 6.9 [3.2–12.9] |
|  | **IL-10/TNF-α** | - | 3.4 | 2.0 [1.1–3.4] |
|  | **IL-10/IL-6** | - | 2.9 | 1.4 [0.6–2.5] |
| STH | **IL-6** | 1.0 [0.8–4.8] | 5.0 [0.8–11.4] | 2.6 [0.8–8.0] |
|  | **TNF-α** | 7.2 [4.6–10.9] | 3.4 [2.0–3.8] | 6.4 [3.4–10.9] |
|  | **IL-10** | 7.1 [6.0–10.7] | 4.7 [3.4–15.2] | 8.3 [4.1–12.9] |
|  | **IL-10/TNF-α** | 1.0 [0.8–1.5] | 2.0 [0.9–4.7] | 1.1 [0.7–2.0] |
|  | **IL-10/IL-6** | 7.1 [4.1–8.8] | 2.9 [0.2–5.8] | 2.1 [1.1–5.5] |
| Intestinal protozoa only | **IL-6** | 13.9 [3.3–119.5] | 5.0 [0.8–7.1] | 3.1 [0.8–6.0] |
|  | **TNF-α** | 4.3 [3.1–5.1] | 5.3 [4.6–6.3] | 5.1 [0.9–5.7] |
|  | **IL-10** | 16.7 [9.7–54.6] | 8.7 [7.1–13.1] | 7.6 [5.7–223.8] |
|  | **IL-10/TNF-α** | 4.8 [2.5–13.0] | 1.6 [1.3–2.8] | 1.5 [1.2–2.1] |
|  | **IL-10/IL-6** | 2.4 [1.0–5.4] | 2.0 [1.2–8.8] | 2.1 [0.7–7.9] |

The IL-10/TNF-α and IL-10/IL-6 ratios had contrasting profiles when compared between older and younger children. Thus, the IL-10/IL-6 ratio was low among the infected older children and high among the youngest children. The IL-10/TNF-α ratio was low in young children with an STH and in older children with intestinal protozoa, while it was high in young children with intestinal protozoa.

This ratio was higher among participants aged 5-to-15 years old (3.4) than among unparasitized participants (2.3 [1.2–25.4]) and lower (1.6 [1.3–2.8]) in groups of subjects with filariasis and intestinal protozoan infection. Adults presented equally high ratios in cases of filariasis (2.0 [1.1–3.4]) and low ratios in cases of geohelminthiasis (1.1 [0.7–2.0]) compared to those of non-infected subjects (1.6 [1.3–1.9]). IL-10/IL-6 was 18-fold (7.1 [4.1–8.8]) and 6-fold (2.4 [1.0–5.4]) higher in younger children with STH and intestinal protozoan infections than in unparasitized children. Among older children, those with filariae (2.9), an STH (2.9 [0.2–5.8]) and intestinal protozoa (2.0 [1.2–8.8]) presented lower IL-10/IL-6 ratios than the control population (5.0 [1.3–8.6]). In adults, this ratio was higher in the case of STH (2.1 [1.1–5.5]) and intestinal protozoan (2.1 [0.7–7.9]) carriage (Table 3).

## Multivariate analysis

Comparison of the IL-6, IL-10 and TNF-alpha levels and of the IL-10/IL-6 and IL-10/TNF-alpha ratios in the 8 different parasitic groups adjusted by age, urbanization and *P. falciparum*

parasitaemia showed no statistically significant differences (Bonferroni test, P = 0.6). Cytokine concentrations and cytokine ratios were similar across the age groups after adjusting for parasitic infections and urbanization.

## Methods

### Ethics statement

This nested study received ethical clearance from "Comité National d'Ethique pour la Recherche" (CNER) of Gabon under the reference PROT No 003/2016/SG/CNE. The protocol and the questionnaire were also approved by the Ministry of Health. The study was explained to the population. A formal written consent was obtained for all participants. For minors, consent was obtained by the parent or guardian. In the informed consent form, it is indicated that the patient or parent/guardian (for child participants) agrees that these samples can be used for other research purposes. All samples were analysed with the approbation of participants during the main study when samples were collected [16].

### Study sites and populations

This study was based on cross-sectional surveys of participants with asymptomatic malaria carried out from September 2013 to June 2016 in five out of the nine provinces of Gabon, with different degrees of urbanization (Fig 4). Samples taken from individuals participating in villages located around the main towns of the Ogooué-Ivindo and Haut-Ogooué provinces and those from patients at the Centre Hospitalier Régional (CHR) d'Oyem, (Woleu-Ntem Province), the CHR de Koula-Moutou and Dienga (Ogooué-lolo Province), the CHR de Melen and the Lalala Public Health Center (Estuaire Province) were used for this study. Lalala is a suburban area of the capital city of Libreville, an urban area; the CHR of Melen is located in a suburban area, and the CHR d'Oyem of Koula-Moutou and villages in Ogooué-Ivindo, Haut-Ogooué and Dienga are located in rural areas.

Inclusion criteria for sample selection included the following: absence of fever (axillary temperature $\leq 37.5˚C$) or absence of history of fever the day of the screening and during the week preceding the consultation, absence of other clinical symptoms suggestive of malaria, absence of antimalarial drug uptake the last two weeks, absence of any other severe medical condition and sickle cell disease, permanent residence in the study area, agreement to fill out the questionnaire and written informed consent.

All plasma controls were those from individuals with no history of or current *P. falciparum* or intestinal parasite infection.

### Procedures for sample selection for the immunological analysis

During the main surveys, patient recruitment, demographic, socioeconomic, environmental and parasitological data were collected in standardized case report forms. Among the 843 patients recruited for the main surveys, 400 provided blood and stool samples.

Then, five groups of samples from participants with different profiles of parasitic infection were constituted: plasma from those with (i) *Plasmodium spp.* infection alone, (ii) helminth infection (blood-borne filariae and intestinal worms), (iii) intestinal protozoan infection alone, (iv) *Plasmodium spp.*/helminth co-infection and (v) *Plasmodium spp.*/intestinal protozoan co-infection. The latter group included samples without detected parasites according to the method used for diagnosis; individuals with no parasites constituted the control group.

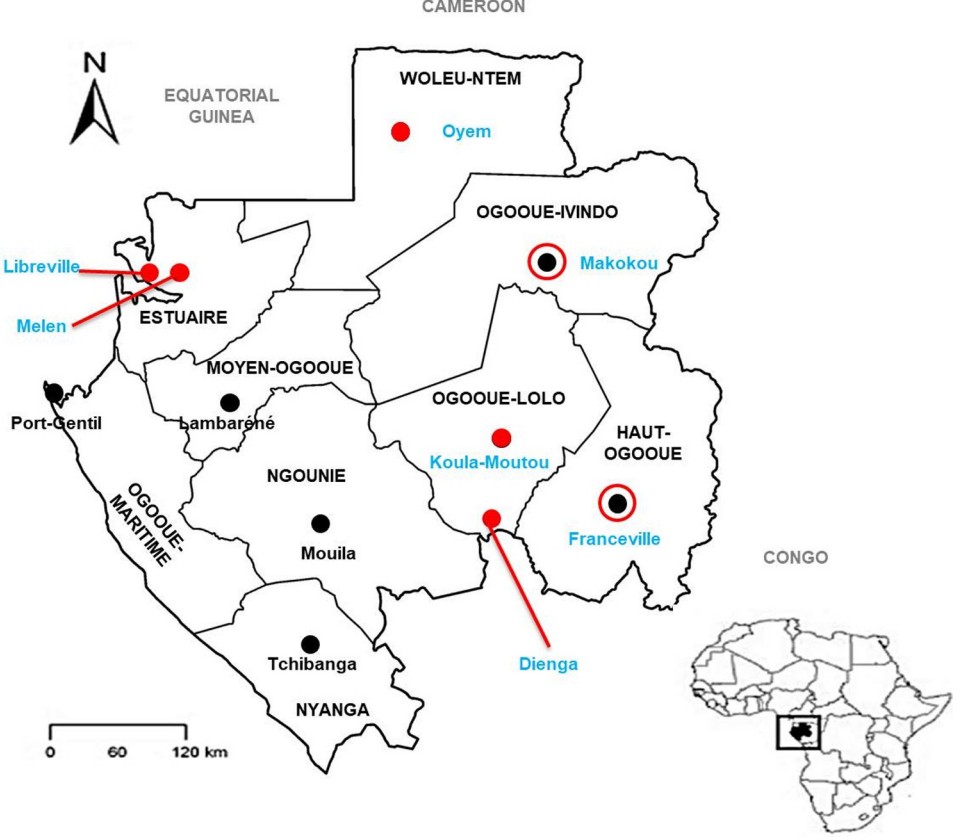

**Fig 4. Study sites where samples were collected.** Map of Gabon with the names of the nine (9) provinces. The black dots on the map correspond to the capitals of each province. Red circles represent many villages around the site, and red full stops indicate a site. Libreville, Oyem and Koula-Moutou are represented in red full stop to indicate study sites but are also capitals of province. Blue indicates the names of sites that patients were recruited from for the present study. Map adapted from: [16].

## Sample processing

After collection, whole blood was used for microscopic malaria and blood filarial diagnosis. After centrifugation, blood pellets were stored at -20˚C for the molecular detection of *P. falciparum*; plasma was directly stored at -20˚C on site, transported to the laboratory in a cooled container and stored at -80˚C prior to immunological analysis. The participants were provided with clean, labelled stool collection pots with clear instructions to ensure that stool samples were collected correctly.

## Parasite microscopic diagnosis

The detection of plasmodial parasites was performed by thin and thick smears using the Lambaréné method [20]. For four individuals, malarial diagnosis was performed with rapid diagnostic tests for malaria. Whole-blood direct microscopic examination and leukoconcentration techniques allowed for the identification of the blood filariae *L. loa* and *M. perstans* [21]. To determine the presence of intestinal parasites, three techniques were performed: direct stool examination, merthiolate-iodine-formaldehyde colouration and parasite culture. These techniques are described in detail elsewhere [16].

For all the techniques for the detection of blood and intestinal parasites, two trained operators were necessary to validate a result. In case of discordance, a third reading was carried out. This procedure is carried out during routine examinations by the technicians of the Laboratory of Parasitology.

## DNA extraction and submicroscopic plasmodial infection

After the microscopic diagnosis of malaria, blood samples of those with negative blood smears were screened for submicroscopic plasmodial infection by nested polymerase chain reaction after DNA extraction.

Precisely, DNA was extracted from peripheral blood collected in EDTA tubes according to the manufacturer's instructions. The 18S rRNA malarial genes were amplified by nested PCR according to the protocol described by Snounou and Singh [22]. Briefly, for the first reaction, the rPLU 1 (5' TCA AAG ATT AAG CCA TGC AAG TGA 3') and 5 (5' CCT GTT GTT GCC TTA AAC TCC 3') primer pair was used. The product generated in this reaction served as a template in the second reaction, performed with the rPLU 3 (5' TTT TTA TAA GGA TAA CTA CGG AAA AGC TGT 3') and 4 (5' TAC CCG TCA TAG CCA TGT TAG GCC AAT ACC 3') primer pair, generating a 235 bp fragment. Visualization of the PCR products was carried out using 2% agarose gel electrophoresis. At each experiment, positive and negative controls were included to validate the obtained results. Participants identified as infected *with Plasmodium sp*. had positive PCR results or positive thick smears.

## Circulating cytokine measurement

A cytometric bead array human cytokine kit from Becton Dickinson (CBA kit, BD Biosciences, San Diego, CA, USA) was used to measure plasma levels of IL-6, IL-10 and TNF-$\alpha$ according to the manufacturer's instructions. Samples were centrifuged, and one volume of the supernatant obtained was diluted into two volumes of the assay diluent. Then, the technique was performed as described by Böstrom and colleagues [23]. The highest standard concentration was 2500 pg/mL, and the lowest was 5 pg/mL. Calibration, sample acquisition in duplicate and the standard were applied using a BD FACSCalibur flow cytometer (FACSCalibur, Becton Dickinson, Le pont de Claix Cedex, France), and the results were analysed with FCAPArray v1.0.1 software (SoftFlow, Pécs, Hungary). The detection limits of cytokines were 1.6 pg/mL, 0.13 pg/mL and 1.2 pg/mL for IL-6, IL-10 and TNF-$\alpha$, respectively. If the cytokine concentration was below the detection limit, a value corresponding to half of the detection limit was assigned to the sample.

## Statistical analysis

Statview version 5.0 (SAS Institute Inc.) was used for statistical analysis, and GraphPad Prism version 5.03 and Statview version 5.0 (SAS Institute Inc.) were used for graphical representations of box plots and scatter columns. Cytokine concentrations were transformed with the $\log_{10}$ function for graphical representation. IL-10/TNF-$\alpha$ and IL-10/IL-6 ratios were calculated for each sample. The ranges of cytokine concentrations, *Plasmodium sp*., *L. loa* and *M. perstans* parasitaemia and eosinophilia did not follow normal distributions, and descriptive analyses of these quantitative variables are presented with medians [interquartile ranges, IQRs: $25^{th}$ - $75^{th}$ quartiles]. Tests for the comparison of cytokines and ratios were performed first using the Kruskal–Wallis test for a global comparison and then the Mann–Whitney U test was used for a comparison of medians of two groups with different infection profiles. Only the Mann–Whitney test for pairwise comparisons with Bonferroni correction was used to determine the significant difference in the cytokine concentrations and cytokine ratios. To perform

multivariate analysis, R software was used to build and evaluate some models. F tests were performed to statistically test the equality of means in different groups according to age, location, and parasitaemia. A $p$ value less than 0.05 was considered significant. Then, a Bonferroni correction test was used for the pairwise comparison of cytokine concentrations adjusted for age group [24]. For each measurement, there was a comparison between 8 different groups with differing parasite profiles. This means that for each measurement, $n(n-1)/2 = 8(8-1)/2 = 28$ independent pairwise comparisons. This number multiplied by the 5 (IL-6, IL-10, TNF-α, IL-10/IL-6 and IL-10/TNF-α) measurements means that $5*28 = 140$ pairwise comparisons were conducted, and the Bonferroni adjusted $p$ value was $0.05/140 = 0.000357$. Pearson's r test was used to evaluate the correlation between the cytokine level/cytokine ratio (variable Y) and age (variable X). Due to the difference in immune response between populations according to urbanization and to avoid confounding factors, areas were split into urban and rural areas, and correlation coefficient values were obtained for each area.

## Definition

Submicroscopic malaria infection was defined as the absence of the parasite after reading thick and thin blood smears but the presence of plasmodial DNA on nested polymerase chain reaction.

## Discussion

This study is the first to analyse the influence of single or multiple parasitic infections on the plasma cytokine profile of Gabonese individuals. The main aim was to determine the relationship between intestinal parasitoses, as well as loiasis, and the plasma concentrations of different cytokines implicated in malaria pathophysiology.

Lower parasitaemia densities were observed among participants carrying co-infections of either *P. falciparum*-intestinal parasites or *P. falciparum*-filariasis than among those with only *P. falciparum*, suggesting an effect of intestinal parasites and filariae on *P. falciparum* multiplication. Lower parasite burdens in the presence of intestinal helminth infections have been reported elsewhere [25,26]. According to these authors, one explanation could be the existence of immune cross-reactivity between intestinal parasites and *Plasmodium sp*. Indeed, both parasites induce a Th2 immune response, and specific IgG3 produced as a result of intestinal parasitic infection could neutralize plasmodial parasites [27]. In the case of protozoan co-infections, no impact of intestinal protozoa on *Plasmodium sp*. parasitaemia has been described.

*P. falciparum*-infected participants, who were asymptomatic when included, had higher median levels of IL-6 and IL-10 than the control participants, but there was no difference in the TNF-α level. In the context of IL-10, this result is consistent with the findings of earlier studies [28,29]. The immune-regulatory and anti-inflammatory cytokine IL-10 is known to downregulate the expression of Th1-type cytokines such as IL-6, TNF, and IL-1 [30]. Here, we found IL-10 to be implicated in the inhibition of the production of TNF-α but not IL-6 during plasmodial infection. It is possible that the downregulation of TNF is correlated with the absence of clinical signs. The TNF-α level did not vary in the control population compared to that in *Plasmodium*-infected participants. TNF-α is the cytokine most implicated in the development of the clinical signs of malaria [31], while IL-10 and IL-6 are present at high levels in patients with symptomatic malaria [32–34]. The asymptomatic status of participants at the time of the study presented here could indicate the acquisition of premunition, which limits the appearance of clinical symptoms, and/or that participants were sampled at an early stage of the infection before progression towards symptomatic disease [35]. It is important to note

that, in the current Gabonese context, younger children are found to be less frequently infected by malaria parasites than older children [16,36], and the number of adults presenting with clinical malaria is increasing, suggesting a loss of malaria premunition. This epidemiological picture most likely primarily reflects the switch, since 2003, to the use of artemisinin combination therapy for malaria in children under five years old.

Previous studies conducted in Gabon showed malaria to be co-endemic with other parasitic infections that are present at a similar or higher prevalence (compared to malaria) and that can alter immunity to malaria [3,4,16,17]. Here, intestinal parasite infections (IPIs) and blood filariasis were associated with decreased Th1 (IL-6) and Treg (IL-10) responses. *Blastocystis sp.*, the most prevalent intestinal parasite, has immunomodulatory effects with the induction of pro-inflammatory cytokine responses [37]. The prevalence of *Blastocystis* has increased across the country [16,17]. Therefore, it was hypothesized that together with its implication in dysbiosis, *Blastocystis* could affect cytokine production and influence *P. falciparum* carriage. A study in Pakistan showed that *Blastocystis sp.* type 1 was associated with low IL-10 production in the blood but, in stool samples, *Blastocystis sp.* generates an anti-inflammatory environment [38,39].

The impact of other pathogenic intestinal protozoa on the cytokine profile was not investigated here because of the low number of infected patients and the absence of single *E. histolytica* and *G. duodenalis* mono-infection or co-infection with *P. falciparum*. However, during intestinal protozoan/*P. falciparum* infection, a significant reduction in pro-inflammatory cytokine levels (IL-6, TNF-α) was observed, suggesting that infection with *G. duodenalis*, *E. histolytica*, or *Blastocystis sp.* decreases the *P. falciparum*-induced Th1 response, thereby contributing to the absence of clinical symptoms. This scenario would also explain the difference in IL-10 levels between participants with intestinal protozoa and those from the control group. The impact of each of these three protozoan parasites as well as the different subtypes of *Blastocystis sp.* on the global and specific cytokine response merits further exploration.

Individuals with a low Th2-Treg/Th1 ratio are less susceptible to malaria but have a greater likelihood of clinical disease when infected [40,41]. Here, we observed that IL-10 and IL-6 levels were 1.3- to 18.0-fold lower in those with malaria-intestinal parasite co-infections (both helminths and protozoa) than in those with plasmodial mono-infections, although they were still higher than those in uninfected participants. A trend towards a downregulation of TNF-α expression was observed in co-infections with intestinal parasites. This regulatory effect on pro-inflammatory cytokine production during helminth and *P. falciparum* co-infections is well-described [9,12]. Elevated IL-10 levels have been observed in patients with *Plasmodium*-schistosomiasis and malaria-hookworm co-infection [12,42,43]. The higher IL-10/TNF-α ratio in the group of participants with *Plasmodium* mono-infection than in the group with *Plasmodium*/helminth co-infection suggests a possible lower risk of developing clinical signs in the case of STH/*P. falciparum* co-infection, as demonstrated by Frosch & John [40]. Additionally, the IL-10/IL-6 ratios suggest that *Plasmodium* and intestinal protozoan co-infected participants would show association with asymptomatic carriage of these parasites. However, a meta-analysis of young African children with helminth-*P. falciparum* co-infections concluded that they are more susceptible to *P. falciparum* infection [44]. Comparative analysis of uninfected unexposed individuals, infected asymptomatic individuals and patients with clinical symptoms is needed. If such analysis confirms the present results, then it can be suggested that exposure to both IPI and malaria in the context of the observed increasing prevalence of intestinal parasites in the country would partly explain the increasing frequency of asymptomatic *P. falciparum* carriage observed in the country [36,45,46].

IL-10/TNF-α ratios show that filaria-infected volunteers more than 5 years old have a higher risk of *P. falciparum* infection than participants infected with other parasites. However,

when the IL-10/TNF-α (except for STH infection) and IL-10/IL-6 ratios were analysed, children with intestinal parasites were found to be more susceptible to *P. falciparum* infection, presenting higher values. Intestinal helminths and filarial infections are associated with higher production of IL-10 and consequently with the inhibition of Th1-type cytokine production [14,47,48]. The high IL-10/TNF-α and IL-10/IL-6 ratios observed here in participants with helminth co-infections are consistent with such observations.

The development of the anti-infection immune response is promoted by repeated exposures and the chronicity or persistence of infections. The immune response was studied according to age in uninfected, mono-infected or polyparasitized participants; it differed according to age and type of parasitism. The negative correlation between cytokine levels and age favours greater stimulation in young children. They are actually more at risk of contracting the studied parasites (with the exception of filariasis) and polyparasitism [16]. However, the cross-sectional design of this study does not allow us to confirm that the measured cytokine levels were baseline. Nevertheless, the fact that participants lived in the study area for several months without significant interventions related to malaria or IPI allows a comparison between the different groups determined according to the presence and the type of infection. Children with STHs had low levels of IL-10, as observed in Kenyan children; depending on their age, they may simply require a longer period of chronic infection to develop the typical Treg-related IL-10-dominated profile [12]. The decrease in the IL-6 level in the case of intestinal protozoans can be related to an immunoregulatory effect induced by these parasites; this hypothesis is confirmed by the high IL-10/TNF-α ratio. The analysis of the IL-10/IL-6 ratio, which shows a different profile than that of each cytokine, would highlight the predominance of the anti-inflammatory cytokine response in young children infected with STHs and intestinal protozoa. In contrast, in adults, the comparable level between infected and uninfected individuals is probably linked to better control of these infections through a protective and mature immune response.

The predominance of a pro-inflammatory cytokine profile in the absence of infection may be linked to the greater susceptibility of young children to severe malaria, which is determined by high IL-6 and TNF responses. This phenomenon could partly explain the predominance of severe forms of malaria in young children, although they remain less frequently infected [36,49]. Indeed, the lower basal cytokine Th1 level in healthy children could indicate their greater susceptibility to plasmodial infection due to a less efficient control of *Plasmodium* multiplication that would quickly reach the pyrogenic threshold. In contrast, the downregulation of IL-10 expression observed in older children would favour less control of parasite multiplication at the beginning of infection and therefore of a higher frequency of parasitaemia in this population than in young people.

The cross-sectional design is one of the main limits of this study. Additionally, plasma cytokines could not be considered specifically induced by any given parasitic infection. Specific and more sensitive techniques, such as stimulation of PBMCs in vitro to assess responses to *P. falciparum*, intestinal parasites and filariae, would be useful in this context. Similarly, next-generation sequencing would provide a global picture of the immune response by measuring the transcript levels of Th1-, Th2- and Treg-related cytokines. The influence of helminths and intestinal protozoa on innate and adaptive responses to *P. falciparum* could thus be revealed. The small sample size in groups with different parasitic infections could also explain the lack of association between the cytokine levels and infection profiles in the multivariate analysis. However, the present results already provide a baseline estimation of the overall in vivo cytokine levels implicated in antiparasite immunity in exposed individuals according to their age.

## Conclusions

This study demonstrates the impacts of blood filariae and intestinal parasites (helminths and protozoa) on the cytokine responses involved in susceptibility to infection with *P. falciparum*. No significant results were found in the shis study may be due to the small size of the study population. STHs would be protective against malaria according to the IL-10/TNF-α ratio, while according to the IL-10/IL-6 ratio, intestinal protozoa may have a detrimental effect. These parasites, which are co-endemic with *Plasmodium falciparum* in Gabon, also seem to alter the susceptibility to *P. falciparum* infection according to age, but there is a need to investigate this hypothesis with a larger sample size. Those more than 5 years old and adults seem to be more at risk of *Plasmodium* sp. infection when co-infected with filariasis, according to the association with a higher IL-10/TNF-α ratio, and when co-infected with intestinal parasites, according to the association with a higher IL-10/IL-6 ratio. This finding suggests that co-endemic parasites can also explain the observed epidemiological transition of malaria, with a shift in at-risk populations. Additional studies involving unexposed uninfected and asymptomatic or clinically ill participants are needed to better understand the dynamic interaction between a specific host immune response and polyparasitism.

## Supporting information

**S1 Table. Parasitological characteristics of the study groups. Differences between median percentages of parasitaemia were determined by the Bonferroni test.** [a], [b] and [c]: Comparisons of median parasitaemia between patients with malaria only *versus* malaria/filariasis co-infection ($p = 0.039$), malaria/STH co-infection ($p = 0.0047$), and malaria/intestinal protozoa co-infection ($p = 0.016$). [d] and [e]: Comparisons of median parasitaemia between patients with malaria/filariasis co-infection *versus* malaria/STH co-infection ($p = 0.4$) and malaria/intestinal protozoa co-infection ($p = 0.6$). [f]: Comparison of median parasitaemia between patients with malaria/STH co-infection *versus* malaria/intestinal protozoan co-infection ($p = 0.3$). [g] and [h]: Comparison of median microfilaremia between patients with filariasis *versus* malaria/filariasis co-infection: $p = 0.7$ for *L. loa* and $p = 0.3$ for *M. perstans*. [i] E. h/d: *E. histolytica/dispar*. (DOCX)

## Acknowledgments

We would like to thank the members of the Red Cross in Libreville and the Ministry of Health for giving permission and logistical support for this study to be conducted. We are also grateful to the study participants.

## Author Contributions

**Conceptualization:** Marielle Karine Bouyou-Akotet.

**Data curation:** Noé Patrick M'Bondoukwé, Kowir Pambou Bello.

**Formal analysis:** Noé Patrick M'Bondoukwé, Kowir Pambou Bello.

**Funding acquisition:** Noé Patrick M'Bondoukwé, Marielle Karine Bouyou-Akotet.

**Investigation:** Noé Patrick M'Bondoukwé, Jacques Mari Ndong Ngomo, Jeanne Vanessa Koumba Lengongo.

**Methodology:** Noé Patrick M'Bondoukwé, Reinne Moutongo, Komi Gbédandé, Jacques Mari Ndong Ngomo, Tatiana Hountohotegbé, Rafiou Adamou, Jeanne Vanessa Koumba Lengongo, Adrian John Frederick Luty.

**Project administration:** Noé Patrick M'Bondoukwé, Marielle Karine Bouyou-Akotet.

**Resources:** Marielle Karine Bouyou-Akotet.

**Software:** Kowir Pambou Bello.

**Supervision:** Marielle Karine Bouyou-Akotet.

**Validation:** Denise Patricia Mawili-Mboumba, Adrian John Frederick Luty, Marielle Karine Bouyou-Akotet.

**Visualization:** Noé Patrick M'Bondoukwé, Kowir Pambou Bello.

**Writing – original draft:** Noé Patrick M'Bondoukwé.

**Writing – review & editing:** Noé Patrick M'Bondoukwé, Denise Patricia Mawili-Mboumba, Marielle Karine Bouyou-Akotet.

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
