## [Decision Letter · Decision Letter 0]

10 Aug 2021

Dear Dr. M'Bondoukwe,

Thank you very much for submitting your manuscript "Circulating IL-6, IL-10, TNF-alpha and IL-10/IL-6, IL-10/TNF-alpha Ratios Profiles of Polyparasitized Individuals in Rural and Urban Areas of Gabon" for consideration at PLOS Neglected Tropical Diseases. As with all papers reviewed by the journal, your manuscript was reviewed by members of the editorial board and by several independent reviewers. In light of the reviews (below this email), we would like to invite the resubmission of a significantly-revised version that takes into account the reviewers' comments. 

There needs to be a thorough revising of the central hypothesis of the study and. addressing the issues raised regarding the statistical analysis.

We cannot make any decision about publication until we have seen the revised manuscript and your response to the reviewers' comments. Your revised manuscript is also likely to be sent to reviewers for further evaluation.

Sincerely,

De'Broski R Herbert

Associate Editor

Ana Rodriguez

Deputy Editor

There needs to be a thorough revising of the central hypothesis of the study and. addressing the issues raised regarding the statistical analysis.

Reviewer's Responses to Questions

**Key Review Criteria Required for Acceptance?**

**Methods**

-Are the objectives of the study clearly articulated with a clear testable hypothesis stated?

-Is the study design appropriate to address the stated objectives?

-Is the population clearly described and appropriate for the hypothesis being tested?

-Is the sample size sufficient to ensure adequate power to address the hypothesis being tested?

-Were correct statistical analysis used to support conclusions?

-Are there concerns about ethical or regulatory requirements being met?

Reviewer #1: -Are the objectives of the study clearly articulated with a clear testable hypothesis stated? YES

-Is the study design appropriate to address the stated objectives? YES

-Is the population clearly described and appropriate for the hypothesis being tested? No, it needs further clarification

-Is the sample size sufficient to ensure adequate power to address the hypothesis being tested? No, but given the characteristics of the work it is enough.

-Were correct statistical analysis used to support conclusions? Yes, although in some occasion it needs further explanation. 

-Are there concerns about ethical or regulatory requirements being met? Yes, important official references are missing. 

Further comments follow:

- Figure 1: This figure needs some remodeling and clarification from authors. The box with the final number of patients selected for the study (n=240) is currently named “patients selected randomly”. This is not true, as the samples were not selected randomly but after exclusion criteria showed right above it. The “molecular diagnosis of malaria” box in the middle should be renamed, as all samples are not diagnosed using molecular techniques. I suggest authors change it to just “diagnosis of malaria”. More importantly, numbers do not match. In theory there are 72+5=77 plasmodium positive samples (as per the diagnosis of malaria box); however, in the last row there are 50+7+7+17=81 samples that have plasmodium (alone or in combination). This mismatch applies to the whole manuscript and table 1 as well. 

- L295-297: This information either needs to be referenced to the appropriate section where they are explained, or directly moved to those sections.

Reviewer #2: The main concern about the study is the lack of a clear hypothesis. This lack is highlighted by the confused conclusion presented by the authors. In addition, the results presented in the manuscript did not show any significant correlation worthy of note.

Line 45, not even all negative blood smears means absence of plasmodial infection that could be evidenced by microscopic evaluation of smears. I really didn't understand what the authors were supposed to mean by “submicroscopic plasmodial infection”. The authors should revise the use of “submicroscopic” terminology trough manuscript (line 76).

Line 51, p=0.09 is correct?

In abstract section, it was not clear what was the main objective of the study. There is not a clear conclusion as well.

**Results**

-Does the analysis presented match the analysis plan?

-Are the results clearly and completely presented?

-Are the figures (Tables, Images) of sufficient quality for clarity?

Reviewer #1: -Does the analysis presented match the analysis plan? YES

-Are the results clearly and completely presented? NO (see below)

-Are the figures (Tables, Images) of sufficient quality for clarity? No, they need improvement, further details and clarification

Further comments:

- Table 2 needs further explanation in the legend and in the text. What are the P values in the last column referred to? In other words, what comparisons were made to obtain the P values shown (all groups together, infected vs uninfected, control vs a particular group…)?

- Figure 2: In order to make the figure more attractive visually and easier for the reader to understand, I suggest authors only show statistical differences between groups. Currently it is packed with information almost overlapping, making it hard to read and interpret. 

- L167-169: Please rephrase, it is hard to understand what authors try to convey in this sentence.

- L213-215: These sentences are confusing and are sending contradictory messages.

Reviewer #2: The results presented in the manuscript did not show any significant correlation worthy of note.

**Conclusions**

-Are the conclusions supported by the data presented?

-Are the limitations of analysis clearly described?

-Do the authors discuss how these data can be helpful to advance our understanding of the topic under study?

-Is public health relevance addressed?

Reviewer #1: -Are the conclusions supported by the data presented? YES

-Are the limitations of analysis clearly described? YES

-Do the authors discuss how these data can be helpful to advance our understanding of the topic under study? YES

-Is public health relevance addressed? YES

No further comments for this section.

Reviewer #2: In conclusion section, the authors stated that STH would be protective against malaria while intestinal protozoa may have a detrimental effect, based only in cytokines levels ratio, however, this is not enough to affirm that. The study not has sufficient data to achieve this conclusion.

Still in conclusion, the authors stated that children with more than 5 years old and adults seem to be more at risk of infection when co-infected with filariasis associated with a higher IL-10/TNF-α ratio, and when co-infected with intestinal parasites associated with a higher IL-10/IL-6 ratio. It is not clear what the authors wanted sated here, but this sentence not make it any sense at all. Maybe the authors would mean that co-infected people should be more susceptible to present clinical symptomatology or bad outcome in case of infection with P. falciparum.

**Editorial and Data Presentation Modifications?**

Reviewer #1: - Author’s summary section needs significant remodeling and English editing. Currently it is hard to understand on account of incomplete sentences, wrong choice of words or the lack of a clear link between ideas in different sentences, among others. For instance, in L80-81 it is not clear what authors try to convey here: helminths are not protozoan parasites but they seem to be considered like that.

- In general, the manuscript needs a thorough and deep English editing, as currently it is hard to follow on many parts due to wrong choice of words, missing or wrong prepositions and incomplete sentences, among others. 

- Figures and tables need some further clarification and other minor modifications (see above)

Reviewer #2: (No Response)

**Summary and General Comments**

Reviewer #1: In this work authors explore the influence of diverse co-infection with asymptomatic patients from Gabon by analyzing cytokine profiles and comparing the different groups composing their samples. Despite being relatively simple, the work is well carried out and conclusions are supported by the results, which can further advance in the knowledge of the devastating disease caused by Plasmodium. In general authors did a good job in all sections although there are many parts that need some editing, further clarification or more details. Importantly, English quality is deficient and a thorough and significant editing is needed before the manuscript can be acceptable for publication, as currently many parts are hard to follow or catch the message authors try to convey. Nevertheless, I believe this work has the potential to be published in PNTD should authors address the issues raised in this revision.

Reviewer #2: In this manuscript “Circulating IL-6, IL-10, TNF-alpha and IL-10/IL-6, IL-10/TNF-alpha Ratios Profiles of Polyparasitized Individuals in Rural and Urban Areas of Gabon”, the authors performed the quantification of cytokine levels of plasma of individuals from rural and urban area from seven sites of Gabon with different co-parasitism. The main concern about the study is the lack of a clear hypothesis. This lack is highlighted by the confused conclusion presented by the authors. In addition, the results presented in the manuscript did not show any significant correlation worthy of note. I explain my concerns in more detail below.

Major

In conclusion section, the authors stated that STH would be protective against malaria while intestinal protozoa may have a detrimental effect, based only in cytokines levels ratio, however, this is not enough to affirm that. The study not has sufficient data to achieve this conclusion.

Still in conclusion, the authors stated that children with more than 5 years old and adults seem to be more at risk of infection when co-infected with filariasis associated with a higher IL-10/TNF-α ratio, and when co-infected with intestinal parasites associated with a higher IL-10/IL-6 ratio. It is not clear what the authors wanted sated here, but this sentence not make it any sense at all. Maybe the authors would mean that co-infected people should be more susceptible to present clinical symptomatology or bad outcome in case of infection with P. falciparum.

Minor

Line 45, not even all negative blood smears means absence of plasmodial infection that could be evidenced by microscopic evaluation of smears. I really didn't understand what the authors were supposed to mean by “submicroscopic plasmodial infection”. The authors should revise the use of “submicroscopic” terminology trough manuscript (line 76).

Line 51, p=0.09 is correct?

In abstract section, it was not clear what was the main objective of the study. There is not a clear conclusion as well.

Sentence in line 66-68 is confuse, please revise it. 

Abstract and the author summary seems no to be in line.

PLOS authors have the option to publish the peer review history of their article (what does this mean?). If published, this will include your full peer review and any attached files.

Reviewer #1: No

Reviewer #2: No
---

## [Decision Letter · Decision Letter 1]

6 Mar 2022

Dear Dr. M'Bondoukwé,

We are pleased to inform you that your manuscript 'Circulating IL-6, IL-10, and TNF-alpha and IL-10/IL-6 and IL-10/TNF-alpha Ratio Profiles of Polyparasitized Individuals in Rural and Urban Areas of Gabon' has been provisionally accepted for publication in PLOS Neglected Tropical Diseases.

Best regards,

De'Broski R Herbert

Associate Editor

Ana Rodriguez

Deputy Editor

This is a nice improvement to and important manuscript

Reviewer's Responses to Questions

**Key Review Criteria Required for Acceptance?**

**Methods**

-Are the objectives of the study clearly articulated with a clear testable hypothesis stated?

-Is the study design appropriate to address the stated objectives?

-Is the population clearly described and appropriate for the hypothesis being tested?

-Is the sample size sufficient to ensure adequate power to address the hypothesis being tested?

-Were correct statistical analysis used to support conclusions?

-Are there concerns about ethical or regulatory requirements being met?

Reviewer #1: (No Response)

Reviewer #2: Methods are well presented, and the population was clearly described in the revised version of the manuscript.

**Results**

-Does the analysis presented match the analysis plan?

-Are the results clearly and completely presented?

-Are the figures (Tables, Images) of sufficient quality for clarity?

Reviewer #1: (No Response)

Reviewer #2: Results are presented properly. The limitation of the study due to the low sample size should be highlighted in the manuscript.

**Conclusions**

-Are the conclusions supported by the data presented?

-Are the limitations of analysis clearly described?

-Do the authors discuss how these data can be helpful to advance our understanding of the topic under study?

-Is public health relevance addressed?

Reviewer #1: (No Response)

Reviewer #2: The conclusion should be clear about the study not showing any significant correlation due to the small sample size. There is an overuse of the term "seems to..." in the conclusion. All these hypotheses should be clearly described as they are, a hypothesis that was not demonstrated in the manuscript, avoiding speculation in the conclusion. The conclusion should be reviewed to demonstrate and highlight the results found in the study.

**Editorial and Data Presentation Modifications?**

Reviewer #1: My only recommendation for authors is still to modify Figure 2 by removing the p values, as the current version is really cumbersome to look at: it is loaded with almost overlapping numbers that makes it a very visually confusing graph. Since no statistical differences were found whatsoever, there is no point, in my opinion, to show the specific p values. If authors still want to describe the values for each comparison I recommend adding a table or doing so in the figure legend.

Reviewer #2: (No Response)

**Summary and General Comments**

Reviewer #1: Authors have addressed all my concerns in a very satisfactory way, and I salute them for their efforts. I feel the new revised version has greatly improved and I do, therefore, recommend its publication.

Reviewer #2: Modification in Summary addressed the initial comments.

PLOS authors have the option to publish the peer review history of their article (what does this mean?). If published, this will include your full peer review and any attached files.

Reviewer #1: **Yes: **David Arranz Solis

Reviewer #2: No

---

## [Editor Report · Acceptance letter]

11 Apr 2022

Dear Dr M'Bondoukwé,

We are delighted to inform you that your manuscript, "Circulating IL-6, IL-10, and TNF-alpha and IL-10/IL-6 and IL-10/TNF-alpha Ratio Profiles of Polyparasitized Individuals in Rural and Urban Areas of Gabon," has been formally accepted for publication in PLOS Neglected Tropical Diseases.

Best regards,

Shaden Kamhawi

co-Editor-in-Chief

Paul Brindley

co-Editor-in-Chief
